# Inherent Weight Normalization in Stochastic Neural Networks

**Georgios Detorakis**
Department of Cognitive Sciences
University of California Irvine
Irvine, CA 92697
gdetorak@uci.edu

**Sourav Dutta**
Department of Electrical Engineering
University of Notre Dame
Notre Dame, IN 46556 USA
sdutta4@nd.edu

**Abhishek Khanna**
Department of Electrical Engineering
University of Notre Dame
Notre Dame, IN 46556 USA
akhanna@nd.edu

**Matthew Jerry**
Department of Electrical Engineering
University of Notre Dame
Notre Dame, IN 46556 USA
mjerry@alumni.nd.edu

**Suman Datta**
Department of Electrical Engineering
University of Notre Dame
Notre Dame, IN 46556 USA
sdatta@nd.edu

**Emre Neftci**
Department of Cognitive Sciences
Department of Computer Science
University of California Irvine
Irvine, CA 92697
eneftci@uci.edu

## Abstract

Multiplicative stochasticity such as Dropout improves the robustness and generalizability of deep neural networks. Here, we further demonstrate that always-on multiplicative stochasticity combined with simple threshold neurons are sufficient operations for deep neural networks. We call such models Neural Sampling Machines (NSM). We find that the probability of activation of the NSM exhibits a self-normalizing property that mirrors Weight Normalization, a previously studied mechanism that fulfills many of the features of Batch Normalization in an online fashion. The normalization of activities during training speeds up convergence by preventing internal covariate shift caused by changes in the input distribution. The always-on stochasticity of the NSM confers the following advantages: the network is identical in the inference and learning phases, making the NSM suitable for online learning, it can exploit stochasticity inherent to a physical substrate such as analog non-volatile memories for in-memory computing, and it is suitable for Monte Carlo sampling, while requiring almost exclusively addition and comparison operations. We demonstrate NSMs on standard classification benchmarks (MNIST and CIFAR) and event-based classification benchmarks (N-MNIST and DVS Gestures). Our results show that NSMs perform comparably or better than conventional artificial neural networks with the same architecture.

## 1 Introduction

Stochasticity is a valuable resource for computations in biological and artificial neural networks [9, 32, 2]. It affects neural networks in many different ways. Some of them are (i) escaping local minima during learning and inference [1], (ii) stochastic regularization in neural networks [21, 52], (iii)

Bayesian inference approximation with Monte Carlo sampling [9, 16], (iv) stochastic facilitation [31], and (v) energy efficiency in computation and communication [28, 19].

In artificial neural networks, multiplicative noise is applied as random variables that multiply network weights or neural activities (*e.g.* Dropout). In the brain, multiplicative noise is apparent in the probabilistic nature of neural activations [19] and their synaptic quantal release [8, 51]. Analog non-volatile memories for in-memory computing such as resistive RAMs, ferroelectric devices or phase-change materials [54, 23, 15] exhibit a wide variety of stochastic behaviors [42, 34, 54, 33], including set/reset variability [3] and random telegraphic noise [4]. In crossbar arrays of non-volatile memory devices designed for vector-matrix multiplication (*e.g.* where weights are stored in the resistive or ferroelectric states), such stochasticity manifests itself as multiplicative noise.

Motivated by the ubiquity of multiplicative noise in the physics of artificial and biological computing substrates, we explore here Neural Sampling Machines (NSMs): a class of neural networks with binary threshold neurons that rely almost exclusively on multiplicative noise as a resource for inference and learning. We highlight a striking self-normalizing effect in the NSM that fulfills a role that is similar to Weight Normalization during learning [47]. This normalizing effect prevents internal covariate shift as with Batch Normalization [22], stabilizes the weights distributions during learning, and confers rejection to common mode fluctuations in the weights of each neuron.

We demonstrate the NSM on a wide variety of classification tasks, including classical benchmarks and neuromorphic, event-based benchmarks. The simplicity of the NSM and its distinct advantages make it an attractive model for hardware implementations using non-volatile memory devices. While stochasticity there is typically viewed as a disadvantage, the NSM has the potential to exploit it. In this case, the forward pass in the NSM simply boils down to weight memory lookups, additions, and comparisons.

## 1.1 Related Work

The NSM is a stochastic neural network with discrete binary units and thus closely related to Binary Neural Networks (BNN). BNNs have the objective of reducing the computational and memory footprint of deep neural networks at run-time [14, 44]. This is achieved by using binary weights and simple activation functions that require only bit-wise operations.

Contrary to BNNs, the NSM is stochastic during both inference and learning. Stochastic neural networks are argued to be useful in learning multi-modal distributions and conditional computations [7, 50] and for encoding uncertainty [16].

Dropout and Dropconnect techniques randomly mask a subset of the neurons and the connections during train-time for regularization and preventing feature co-adaptation [21, 52]. These techniques continue to be used for training modern deep networks. Dropout during inference time can be viewed as approximate Bayesian inference in deep Gaussian processes [16], and this technique is referred to as Monte Carlo (MC) Dropout. NSMs are closely related to MC Dropout, with the exception that the activation function is stochastic and the neurons are binary. Similarly to MC Dropout, the "always-on" stochasticity of NSMs can be in principle articulated as a MC integration over an equivalent Gaussian process posterior approximation, fitting the predictive mean and variance of the data. MC Dropout can be used for active learning in deep neural networks, whereby a learner selects or influences the training dataset in a way that optimally minimizes a learning criterion [16, 12].

Taken together, NSM can be viewed as a combination of stochastic neural networks, Dropout and BNNs. While stochastic activations in the binarization function are argued to be inefficient due to the generation of random bits, stochasticity in the NSM, however, requires only one random bit per pass per neuron or per connection. A different approach for achieving zero mean and unit variance is the self-normalizing neural networks proposed in [25]. There, an activation function in non-binary, deterministic networks is constructed mathematically so that outputs are normalized. In contrast, in the NSM unit, normalization in the sense of [47] emerges from the multiplicative noise as a by-product of the central limit theorem. This establishes a connection between exploiting the physics of hardware systems and recent deep learning techniques, while achieving good accuracy on benchmark classification tasks. Such a connection is highly significant for the devices community, as it implies a simple circuit (threshold operations and crossbars) that can exploit (rather than mitigate) device non-idealities such as read stochasticity.

In recurrent neural networks, stochastic synapses were shown to behave as stochastic counterparts of Hopfield networks [38], but where stochasticity is caused by multiplicative noise at the synapses (rather than logistic noise in Boltzmann machines). These were shown to surpass the performances of equivalent machine learning algorithms [20, 36] on certain benchmark tasks.

## 1.2 Our Contribution

In this article, we demonstrate multi-layer and convolutional neural networks employing NSM layers on GPU simulations, and compare with their equivalent deterministic neural networks. We articulate NSM's self-normalizing effect as a statistical equivalent of Weight Normalization. Our results indicate that a neuron model equipped with a hard-firing threshold (*i.e.*, a Perceptron) and stochastic neurons and synapses:

- Is a sufficient resource for stochastic, binary deep neural networks.
- Naturally performs weight normalization.
- Can outperform standard artificial neural networks of comparable size.

The always-on stochasticity gives the NSM distinct advantages compared to traditional deep neural networks or binary neural networks: The shared forward passes for training and inference in NSM are consistent with the requirement of online learning since an NSM implements weight normalization, which is not based on batches [47]. This enables simple implementations of neural networks with emerging devices. Additionally, we show that the NSM provides robustness to fluctuations and fixed precision of the weights during learning.

## 1.3 Applications

During inference, the binary nature of the NSM equipped with blank-out noise makes it largely multiplication-free. As with the Binarynet [13] or XNORnet [44], we speculate that they can be most advantageous in terms of energy efficiency on dedicated hardware.

The NSM is of interest for hardware implementations in memristive crossbar arrays, as threshold units are straightforward to implement in CMOS and binary inputs mitigate read and write non-idealities in emerging non-volatile memory devices while reducing communication bandwidth [54]. Furthermore, multiplicative stochasticity in the NSM is consistent with the stochastic properties of emerging nanodevices [42, 34]. Exploiting the physics of nanodevices for generating stochasticity can lead to significant improvements in embedded, dedicated deep learning machines.

## 2 Methods

### 2.1 Neural Sampling Machines (NSM)

We formulate the NSM as a stochastic neural network model that exploits the properties of multiplicative noise to perform inference and learning. For mathematical tractability, we focus on threshold (sign) units, where $\text{sgn} : \mathbb{R} \to [-1, 1]$,

$$z_i = \text{sgn}(u_i) = \begin{cases} 1 & \text{if } u_i \geq 0 \\ -1 & \text{if } u_i < 0 \end{cases}, \tag{1}$$

where $u_i$ is the pre-activation of neuron $i$ given by the following equation

$$u_i = \sum_{j=1}^{N} \xi_{ij} w_{ij} z_j + b_i + \eta_i, \tag{2}$$

where $\xi_{ij}$ and $\eta_i$ represent multiplicative and additive noise terms, respectively. Both $\xi$ and $\eta$ are independent and identically distributed (*iid*) random variables. $w_{ij}$ is the weight of the connection between neurons $i$ and $j$, $b_i$ is a bias term, and $N$ is the number of input connections (fan-in) to neuron $i$. Note that multiplicative noise can be introduced at the synapse ($\xi_{ij}$), or at the neuron ($\xi_i$).

Since the neuron is a threshold unit, it follows that $P(z_i = 1|\mathbf{z}) = P(u_i \geq 0|\mathbf{z})$. Thus, the probability that unit $i$ is active given the network state is equal to one minus the cumulative distribution function

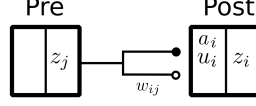

Figure 1: **Blank-out synapse with scaling factors.** Weights are accumulated on $u_i$ as a sum of a deterministic term scaled by $\alpha_i$ (filled discs) and a stochastic term with fixed blank-out probability $p$ (empty discs).

of $u_i$. Assuming independent random variables $u_i$, the central limit theorem indicates that the probability of the neuron firing is $P(z_i = 1|\mathbf{z}) = 1 - \Phi(u_i|\mathbf{z})$ (where $\Phi$ is the cumulative distribution function of normal distribution) and more precisely

$$P(z_i = 1|\mathbf{z}) = \frac{1}{2}\left(1 + \text{erf}\left(\frac{\mathbb{E}(u_i|\mathbf{z})}{\sqrt{2\text{Var}(u_i|\mathbf{z})}}\right)\right), \tag{3}$$

where $\mathbb{E}(u_i)$ and $\text{Var}(u_i)$ are the expectation and variance of state $u_i$.

In the case where only independent additive noise is present, equation (2) is rewritten as $u_i = \sum_{j=1}^N w_{ij}z_j + b_i + \eta_i$ and the expectation and variance are given by $\mathbb{E}(u_i|\mathbf{z}) = \sum_{j=1}^N w_{ij}z_j + b_i + \mathbb{E}(\eta)$ and $\text{Var}(u_i|\mathbf{z}) = \text{Var}(\eta)$, respectively. In this case, equation (3) is a sigmoidal neuron with an $\text{erf}$ activation function with constant bias $\mathbb{E}(\eta)$ and constant slope $\text{Var}(\eta)$. Thus, besides the sigmoidal activation function, the additive noise case does not endow the network with any extra properties.

In the case of multiplicative noise, equation (2) becomes $u_i = \sum_{j=1}^N \xi_{ij}w_{ij}z_j + b_i$ and its expectation and variance are given by $\mathbb{E}(u_i|\mathbf{z}) = \mathbb{E}(\xi)\sum_{j=1}^N w_{ij}z_j$ and $\text{Var}(u_i|\mathbf{z}) = \text{Var}(\xi)\sum_{j=1}^N w_{ij}^2$, respectively. In this derivation, we have used the fact that the square of a sign function is a constant function ($\text{sgn}^2(x) = 1$). In contrast to the additive noise case, $\text{Var}(u_i|\mathbf{z})$ is proportional to the square of the input weight parameters. The probability of neurons being active becomes:

$$P(z_i = 1|\mathbf{z}) = \frac{1}{2}\left(1 + \text{erf}\left(\mathbf{v}_i \cdot \mathbf{z}\right)\right),$$
$$\text{with } \mathbf{v}_i = \beta_i \frac{\mathbf{w}_i}{||\mathbf{w}_i||_2}, \tag{4}$$

where $\beta_i$ here is a variable that captures the parameters of the noise process $\xi_i$. In the denominator, we have used the identity $\sqrt{\sum_j w_{ij}^2} = \sqrt{\sum_j w_{ij}^2 z_j^2} = ||\mathbf{w}_i||_2$, where $||\cdot||_2$ denotes the $L2$ norm of the weights of neuron $i$. This term has a normalizing effect on the activation function, similar to weight normalization, as discussed below. Note that the self-normalizing effect is not specific to the distribution of the chosen random variables, and holds as long as the random variables are *iid*.

One consequence of multiplicative noise here is that any positive scaling factor applied to $\mathbf{w}_i$ is canceled out by the norm. To counter this problem and control $\beta_i$ without changing the distribution governing $\xi$, the NSM introduces a factor $a_i$ in the preactivation's equation:

$$u_i = \sum_{j=1}^N (\xi_{ij} + a_i)w_{ij}z_j + b_i. \tag{5}$$

Thanks to the binary nature of $z_i$, equation (5) is multiplication-free except for the term involving $a_i$. Since $a_i$ is defined per neuron, the multiplication operation is only performed once per neuron and time step. In this article, we focus on two relevant cases of noise: Gaussian noise with mean 1 and variance $\sigma^2$, $\xi_{ij} \sim \mathcal{N}(1, \sigma^2)$ and Bernoulli (blank-out) noise $\xi_{ij} \sim Bernoulli(p)$, with parameter $p$. From now on we focus only on the multiplicative noise case.

**Gaussian Noise** In the case of multiplicative Gaussian Noise, $\xi$ in equation (5) is a Gaussian random variable $\xi \sim \mathcal{N}(1, \sigma^2)$. This means that the expectation and variance are $\mathbb{E}(u_i|\mathbf{z}) = (1+a_i)\sum_j w_{ij}z_j$ and $\text{Var}(u_i|\mathbf{z}) = \sigma^2 \sum_j w_{ij}^2$, respectively. And hence, $\beta_i = \frac{1+a_i}{\sqrt{2\sigma^2}}$.

**Bernoulli (Blank-out) Noise** Bernoulli ("Blank-out") noise can be interpreted as a Dropout mask on the neurons or a Dropconnect mask on the synaptic weights (see Fig 1), where $\xi_{ij} \in [0, 1]$ in

equation (5) becomes a Bernoulli random variable with parameter $p$. Since the $\xi_{ij}$ are independent, for a given $\mathbf{z}$, a sufficiently large fan-in, and $0 < p < 1$, the sums in equation (5) are Gaussian-distributed with means and variances $\mathbb{E}(u_i|\mathbf{z}) = (p+a_i)\sum_j w_{ij}z_j$ and $\text{Var}(u_i|\mathbf{z}) = p(1-p)\sum_j w_{ij}^2$, respectively. Therefore we obtain: $\beta_i = \frac{p+a_i}{\sqrt{2p(1-p)}}$.

We observed empirically that whether the neuron is stochastic or the synapse is stochastic did not significantly affect the results.

## 2.2 NSMs implements Weight Normalization

The key idea in weight normalization [47] is to normalize unit activity by reparameterizing the weight vectors. The reparameterization used there has the form: $\mathbf{v}_i = \beta_i \frac{\mathbf{w}_i}{||\mathbf{w}_i||}$. This is exactly the form obtained by introducing multiplicative noise in neurons (equation (4)), suggesting that NSMs inherently perform weight normalization in the sense of [47]. The authors argue that decoupling the magnitude and the direction of the weight vectors speeds up convergence and confers many of the features of batch normalization. To achieve weight normalization effectively, gradient descent is performed with respect to the scalars $\beta$ (which are themselves parameterized with $a_i$) in addition to the weights $\mathbf{w}$:

$$\partial_{\beta_i}\mathcal{L} = \frac{\sum_j w_{ij}\partial_{v_{ij}}\mathcal{L}}{||\mathbf{w}_i||} \tag{6}$$

$$\partial_{w_{ij}}\mathcal{L} = \frac{\beta_i}{||\mathbf{w}_i||}\partial_{v_{ij}}\mathcal{L} - \frac{\mathbf{w}_i\beta_i}{||\mathbf{w}_i||^2}\partial_{\beta_i}\mathcal{L} \tag{7}$$

## 2.3 NSM Training Procedure

Neural sampling machines (and stochastic neural networks in general) are challenging to train because errors cannot be directly back-propagated through stochastic nodes. This difficulty is compounded by the fact that the neuron state is a discrete random variable, and as such the standard reparametrization trick is not directly applicable [17]. Under these circumstances, unbiased estimators resort to minimizing expected costs through the family of REINFORCE algorithms [53, 7, 35] (also called score function estimator and likelihood-ratio estimator). Such algorithms have general applicability but gradient estimators have impractically high variance and require multiple passes in the network to estimate them [43]. Straight-through estimators ignore the non-linearity altogether [7], but result in networks with low performance. Several work have introduced methods to overcome this issue, such as in discrete variational autoencoders [46], bias reduction techniques for the REINFORCE algorithm [18] and concrete distribution approach (smooth relaxations of discrete random variables) [30] or other reparameterization tricks [48].

Striving for simplicity, here we propagate gradients through the neurons' activation probability function. This approach theoretically comes at a cost in accuracy because the rule is a biased estimate of the gradient of the loss. This is because the gradients are estimated using activation probability. However, it is more efficient than REINFORCE algorithms as it uses the information provided by the gradient back-propagation algorithm. In practice, we find that, provided adequate initialization, the gradients are well behaved and yield good performance while being able to leverage existing automatic differentiation capabilities of software libraries (*e.g.* gradients in Pytorch [40]). In the implementation of NSMs, probabilities are only computed for the gradients in the backward pass, while only binary states are propagated in the forward pass (see SI 4.2).

To assess the impact of this bias, we compare the above training method with Concrete Relaxation which is unbiased [30]. The NSM network is compatible with the binary case of Concrete relaxation. We trained the NSM using BinConcrete units on MNIST data set (Test Error Rate: $0.78\%$), and observed that the angles between the gradients of the proposed NSM and BinConcrete are close (see SI 4.11).

Unless otherwise stated, and similarly to [47], we use a data-dependent initialization of the magnitude parameters $\beta$ and the bias parameters over one batch of 100 training samples such that the

Table 1: Classification error on the permutation invariant MNIST task (test set). Error is estimated by averaging test errors over 100 samples (for NSMs) and over the 50 last epochs.

| Data set | Network | NSM |
|---|---|---|
| PI MNIST | NSM 784–300–300–300–10 | 1.36 % |
| PI MNIST | StNN 784–300–300–300–10 | 1.47 % |
| PI MNIST | NSM scaled 784–300–300–300–10 | 1.38 % |

preactivations to each layer have zero mean and unit variance over that batch:

$$\beta \leftarrow \frac{1}{\sigma}, \qquad\qquad b \leftarrow -\frac{\mu||w||\sqrt{2Var(\xi)}}{\sigma}, \qquad (8)$$

where $\mu$ and $\sigma$ are feature-wise means and standard deviations estimated over the minibatch. For all classification experiments, we used cross-entropy loss $\mathcal{L}^n = -\sum_i t_i^n \log p_i^n$, where $n$ indexes the data sample and $p_i$ is the Softmax output. All simulations were performed using Pytorch [40]. All NSM layers were built as custom Pytorch layers (for more details about simulations see SI 4.8).[1]

## 3 Experiments

### 3.1 Multi-layer NSM Outperforms Standard Stochastic Neural Networks in Speed and Accuracy

In order to characterize the classification abilities of the NSM we trained a fully connected network on the MNIST handwritten digit image database for digit classification. The network consisted of three fully-connected layers of size 300, and a Softmax layer for 10-way classification and all Bernoulli process parameters were set to $p = .5$. The NSM was trained using back-propagation and a softmax layer with cross-entropy loss and minibatches of size 100. As a baseline for comparison, we used the stochastic neural network (StNN) presented in [27] without biases, with a sigmoid activation probability $P_{sig}(z_i = 1|\mathbf{z}) = \text{sigmoid}(\mathbf{w}_i \cdot \mathbf{z})$.

The results of this experiment are shown in Table 1. The 15th, 50th and 85th percentiles of the input distributions to the last hidden layer during training is shown in Fig. 2. The evolution of the distribution in the NSM case is more stable, suggesting that NSMs indeed prevent internal covariate shift.

Both the speed of convergence and accuracy within 200 iterations are higher in the NSM compared to the StNN. The higher performance in the NSM is achieved using inference dynamics that are simpler than the StNN (sign activation function compared to a sigmoid activation function) and using binary random variables.

### 3.2 Robustness to Weight Fluctuations

The decoupling of the weight matrix as in $\mathbf{v}_i = \beta_i \frac{\mathbf{w}_i}{||\mathbf{w}_i||}$ introduces several additional advantages in learning machines. During learning, the distribution of the weights for a layer tend to remain more stable in NSM compared to the StNN (SI Fig. 4). This feature can be exploited to mitigate saturation at the boundaries of fixed range weight representations (*e.g.* in fixed-point representations or memristors). Another subtle advantage from an implementation point of view is that the probabilities are invariant to positive scaling of the weights, *i.e.* $\frac{\alpha \mathbf{w}_i}{||\alpha \mathbf{w}_i||} = \frac{\mathbf{w}_i}{||\mathbf{w}_i||}$ for $\alpha \geq 0$. Table 1 shows that NSM with weights multiplied by a constant factor .1 (called NSM scaled in the table) during inference did not significantly affect the classification accuracy. This suggests that the NSM can be robust to common mode fluctuations that may affect the rows of the weight matrix. Note that this property does not hold for ANNs with standard activation functions (relu, sigmoid, tanh), and the network performance is lost by such scaling (for more details see SI 4.5).

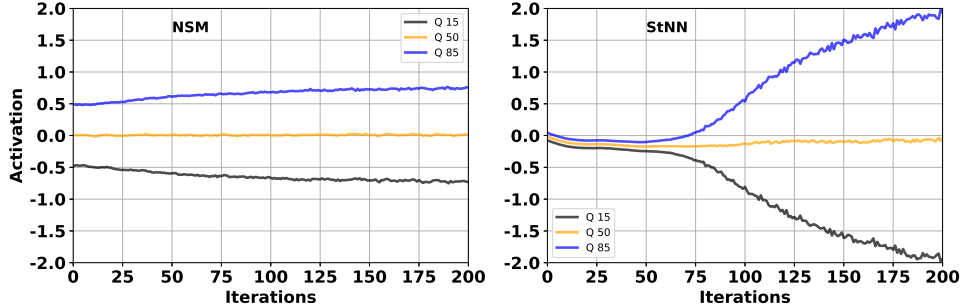

Figure 2: NSM mitigates internal covariate shift. 15th, 50th and 85th percentiles of the input distribution to the last hidden layer (similarly to Fig. 1 in [22]). The internal covariate shift is visible in the StNN as the input distributions change significantly during the learning. The self normalizing effect in NSM performs weight normalization, which is known to mitigate this shift and speed up learning. Each iteration corresponds to one mini-batch update (100 data samples per mini-batch, 20000 data samples total).

### 3.3 Supervised Classification Experiments: MNIST Variants

We validate the effectiveness of NSMs in supervised classification experiments on MNIST [26], EMNIST [11], N-MNIST [39], and DVS Gestures data sets (See Methods) using convolutional architecture. For all data sets, the inputs were converted to $-1/+1$ binary in a deterministic fashion using the function defined in equation (1). For the MNIST variants we trained all the networks for 200 epochs presenting at each epoch the entire dataset. For testing the accuracy of the networks we used the entire test dataset sampling each minibatch 100 times.

NSM models with Gaussian noise (gNSM) and Bernouilli noise (bNSM) converged to similar or better accuracy compared to the architecturally equivalent deterministic models. The results for MNIST, EMNIST and N-MNIST are given in Table 2, where we compare with the deterministic counterpart convolutional neural network (see Table 6 in the SI). In addition we compared with a binary (sign non-linearity) deterministic network (BD), a binary deterministic network with weight normalization (wBD), a stochastic network (noisy rectifier [7]) (SN), and a deterministic binary network (BN). We trained the first three networks using a Straight-Through Estimator [7] (STE) and the latter one using erf function in the backward pass only (*i.e.*, gradients computed on the erf function). The architecture of all these four networks is the same as the NSM's. The training process was the same as for the NSM networks and the results are given in Table 2. From these results we can conclude that NSM training procedure provides better performance than the STE and normalization of binary deterministic networks trained with a similar way as NSM (*e.g.*, BN).

### 3.4 Supervised Classification Experiments: CIFAR10/100

We tested the NSM on the CIFAR10 and CIFAR100 dataset of natural images. We used the model architecture described in [47] and added an extra input convolutional layer to convert RGB intensities into binary values. The NSM non-linearities are sign functions given by equation (1). We used the Adam [24] optimizer, with initial learning rate $0.0003$ and we trained for 200 epochs using a batch size of 100 over the entire CIFAR10/100 data sets ($50K/10K$ images for training and testing respectively). The test error was computed after each epoch and by running 100 times each batch (MC samples) with different seeds. Thus classification was made on the average over the MC samples. After 100 epochs we started decaying the learning rate linearly and we changed the first moment from $0.9$ to $0.5$. The results are given in Table 5. For the NSM networks we tried two different types of initialization. First, we initialized the weights with the values of the already trained deterministic network weights. Second and in order to verify that the initialization does not affect dramatically the training, we initialized the NSM without using any pre-trained weights. In both cases the performance of the NSM was similar as it is indicated in Table 5. We compared with the counterpart deterministic implementation using the exact same parameters and same additional input convolutional layer.

Table 2: (Top) Classification error on MNIST datasets. Error is estimated by averaging test errors over 100 samples (for NSMs), 5 runs, and over the 10 last epochs. Prefix, d-deterministic, b-Bernouilli, g-Gaussian. (Bottom) Comparison of networks on MNIST classification task. The NSM variations Bernoulli (bNSM) and Gaussian (gNSM) are compared with an NSM trained with a Straight-Through Estimator instead of the proposed training algorithm, a deterministic binary (sign non-linearity) network (BD), a BD with weight normalization enabled (wBD), a stochastic network (noisy rectifier) (SN) and a binary network (BN). For more details see section 3.3 in the main text.

| Dataset | dCNN | bNSM | gNSM |
|---|---|---|---|
| MNIST | 0.880% | 0.775 % | 0.805% |
| EMNIST | 6.938% | 6.185 % | 6.256% |
| NMNIST | 0.927% | 0.689 % | 0.701% |

| Model | bNSM | gNSM | bNSM (STE) | BD | wBD | SN | BN |
|---|---|---|---|---|---|---|---|
| Error | 0.775 | 0.805 | 2.13 | 3.11 | 2.72 | 2.05 | 1.10 |

Table 3: Classification error on CIFAR10/CIFAR100. Error is estimated by sampling 100 times each mini-batch (MC samples) and finally averaging over all 100 samples (for NSMs), 5 runs and over the 10 last epochs. Prefix, d-deterministic, b-blank-out, g-Gaussian. The * indicates a network that has not been initialized with pre-trained weights (see main text).

| Dataset | Model | Error |
|---|---|---|
| CIFAR10/100 | bNSM | 9.98% / 34.85% |
| CIFAR10/100 | gNSM | 10.35% / 34.84% |
| CIFAR10/100 | dCNN | 10.47% / 34.37% |
| CIFAR10/100 | bNSM* | 9.94% / 35.19% |
| CIFAR10/100 | gNSM* | 9.81% / 34.93% |

Table 4: Classification error on DVS Gestures data set. Error is estimated by averaging test errors over 100 samples and over the 10 last epochs. Prefix, d-deterministic, b-blank-out, g-Gaussian.

| Dataset | Model | Error |
|---|---|---|
| DVS Gestures | IBM EEDN | 8.23% |
| DVS Gestures | bNSM | 8.56% |
| DVS Gestures | gNSM | 8.83% |
| DVS Gestures | dCNN | 9.16% |

### 3.5 Supervised Classification Experiments: DVS Gestures

Binary neural networks, such as the NSM are particularly suitable for discrete or binary data. Neuromorphic sensors such as Dynamic Vision Sensors (DVS) that output streams of events fall into this category and can transduce visual or auditory spatiotemporal patterns into parallel, microsecond-precise streams of events [29].

Amir *et al.* recorded DVS Gesture data set using a Dynamical Vision Sensor (DVS), comprising 1342 instances of a set of 11 hand and arm gestures, collected from 29 subjects under 3 different lighting conditions. Unlike standard imagers, the DVS records streams of events that signal the temporal intensity changes at each of its $128 \times 128$ pixels. The unique features of each gesture are embedded in the stream of events. To process these streams, we closely follow the pre-processing in [5], where event streams were downsized to $64 \times 64$ and binned in frames of $16$ms. The input of the neural was formed by 6 frames (channels) and only ON (positive polarity) events were used. Similarly to [5], 23 subjects are used for the training set, and the remaining 6 subjects are reserved for testing. We note that the network used in this work is much smaller than the one used in [5].

We adapted a model based on the all convolutional networks of [49]. Compared to the original model, our adaptation includes an additional group of three convolutions and one pooling layer to account for the larger image size compared to the CIFAR10 data set used in [49] and a number of output classes that matches those of the DVS Gestures data set (11 classes). See SI Tab. 7 for a detailed listing of the layers. We trained the network for 200 epochs using a batch size 100. For the NSM network we initialized the weights using the converged weights of the deterministic network. This makes learning more robust and causes a faster convergence.

We find that the smaller models of [49] (in terms of layers and number of neurons) are faster to train and perform equally well when executed on GPU compared to the EEDN used in [5]. The models reported in Amir *et al.* were optimized for implementation in digital neuromorphic hardware, which strongly constrains weights, connectivity and neural activation functions in favor of energetic efficiency.

## 4 Conclusions

Stochasticity is a powerful mechanism for improving the computational features of neural networks, including regularization and Monte Carlo sampling. This work builds on the regularization effect of stochasticity in neural networks, and demonstrates that it naturally induces a normalizing effect on the activation function. Normalization is a powerful feature used in most modern deep neural networks [22, 45, 47], and mitigates internal covariate shift. Interestingly, this normalization effect may provide an alternative mechanism for divisive normalization in biological neural networks [10].

Our results demonstrate that NSMs can (i) outperform standard stochastic networks on standard machine learning benchmarks on convergence speed and accuracy, and (ii) perform close to deterministic feed-forward networks when data is of discrete nature. This is achieved using strictly simpler inference dynamics, that are well suited for emerging nanodevices, and argue strongly in favor of *exploiting* stochasticity in the devices for deep learning. Several implementation advantages accrue from this approach: it is an online alternative to batch normalization and dropout, it mitigates saturation at the boundaries of fixed range weight representations, and it confers robustness against certain spurious fluctuations affecting the rows of the weight matrix.

Although feed-forward passes in networks can be implemented free of multiplications, the weight update rule is more involved as it requires multiplications, calculating the row-wise $L2$-norms of the weight matrices, and the derivatives of the erf function. However, these terms are shared for all connections fanning into a neuron, such that the overhead in computing them is reasonably small. Furthermore, based on existing work, we speculate that approximating the learning rule either by hand [37] or automatically [6] can lead to near-optimal learning performances while being implemented with simple primitives.

## Footnotes

[1]`https://github.com/nmi-lab/neural_sampling_machines`

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
