[Supplementary Material]

# Supplementary Information

## 4.1 Table of Abbreviations

Table 5: Abbreviations used in the main text and in the SI.

| Abbreviation | Definition |
|---|---|
| NSM | Neural Sampling Machine |
| b/gNSM | Bernoulli/Gaussian NSM |
| S2M | Synaptic Sampling Machine |
| StNN | Stochastic Neural Network |
| BNN | Binary Neural Networks |
| DVS | Dynamic Vision Sensor |
| BD | Deterministic Binary Network with sgn as non-linearity |
| wBD | Same as BD with weight normalization enabled |
| SN | Stochastic network with noisy rectifier |
| BN | Binary network |
| STE | Straight-Through Estimator |

## 4.2 Computation of Gradients in NSM Computational Graph

Figure 3: Gradient estimation in NSM computation graph. For the NSM network the gradient $\nabla_\theta L(x)$ is computed via back-propagation on the probability $P_\theta(\mathbf{z})$ only in the backward pass (see equation (3) and main text). The light-green node indicates a stochastic discrete node that propagates the activity of units to the next layer only in the forward pass. The parameters here are $\boldsymbol{\theta} = (\mathbf{w}, \boldsymbol{\beta})$ (see main text).

## 4.3 NSM in Convolutional Neural Networks

CNN perform state-of-the-art in several visual, auditory and natural language tasks by assuming prior structure to the connectivity and the weight matrices [26, 17]. The NSM with stochastic neurons can be similarly extended to the convolution operation as follows (bias parameters omitted):

$$u_{ijk} = Conv(\mathbf{w}_k, \mathbf{z})$$
$$= \sum_{q=1}^{Q} \sum_{m=1}^{H} \sum_{n=1}^{V} (\xi_{i+m,j+n,q} + a_{ijk}) w_{m,n,q,k} z_{i+m,j+n,q} \tag{9}$$

where $Q$ is the number of input channels and $H, V$ are height and width of the filter, respectively. In the case of neural stochasticity, existing software libraries of the convolution can be used. In contrast, synaptic stochasticity, requires modification of such libraries due to the sharing of the filter parameters. While it is possible to do so, we have not observed significant differences in using neural or synaptic stochasticity. Therefore only neural stochasticity is used for convolution operations. Similarly to the case without convolutions, the activation probability becomes:

$$P(z_{ijk} = 1|\mathbf{z})\frac{1}{2}\left(1 + \mathrm{erf}\left(Conv(\mathbf{v}_k, \mathbf{z})\right)\right),$$

$$\text{with } \mathbf{v}_i = \beta_{ijk}\frac{\mathbf{w}_k}{||\mathbf{w}_k||}, \tag{10}$$

where,

$$||\mathbf{w}_k|| = \sqrt{\sum_{m=1}^{H}\sum_{n=1}^{V}\sum_{q=1}^{Q} w_{m,n,q,k}^2}. \tag{11}$$

## 4.4 Derivation of Gradients (Equations (6) and (7))

In this section we derive equations (6) and (7). Therefore, if we differentiate through $\mathbf{v}_i = \beta_i\frac{\mathbf{w}_i}{||\mathbf{w}_i||}$, we obtain equation (6) from

$$\frac{\partial\mathcal{L}}{\partial\boldsymbol{\beta}_i} = \frac{\partial\mathcal{L}}{\partial\mathbf{v}_i}\frac{\partial\mathbf{v}_i}{\partial\boldsymbol{\beta}_i}$$

$$= \frac{\partial\mathcal{L}}{\partial\mathbf{v}_i}\frac{\mathbf{w}_i}{||\mathbf{w}_i||}$$

$$= \frac{\sum_j w_{ij}\partial_{v_{ij}}\mathcal{L}}{||\mathbf{w}_i||}. \tag{12}$$

And it is obvious that equation (6) is equation (12). For obtaining equation (7) we have

$$\frac{\partial\mathcal{L}}{\partial\mathbf{w}_i} = \frac{\partial\mathcal{L}}{\partial\mathbf{v}_i}\frac{\partial\mathbf{v}_i}{\partial\mathbf{w}_i}$$

$$= \frac{\partial\mathcal{L}}{\partial\mathbf{v}_i}\frac{\beta_i\partial\frac{\mathbf{w}_i}{||\mathbf{w}_i||}}{\partial\mathbf{w}_i}$$

$$= \frac{\partial\mathcal{L}}{\partial\mathbf{v}_i}\left(\beta_i\frac{\frac{\partial\mathbf{w}_i}{\partial\mathbf{w}_i}||\mathbf{w}_i|| - \mathbf{w}_i\frac{\partial||\mathbf{w}_i||}{\partial\mathbf{w}_i}}{||\mathbf{w}_i||^2}\right)$$

$$= \frac{\partial\mathcal{L}}{\partial\mathbf{v}_i}\left(\beta_i\frac{||\mathbf{w}_i|| - \mathbf{w}_i\frac{\sum_j w_{ij}}{||\mathbf{w}_i||}}{||\mathbf{w}_i||^2}\right)$$

$$= \frac{\partial\mathcal{L}}{\partial\mathbf{v}_i}\frac{\beta_i||\mathbf{w}_i||}{||\mathbf{w}_i||^2} - \frac{\partial\mathcal{L}}{\partial\mathbf{v}_i}\frac{\beta_i\mathbf{w}_i\sum_j w_{ij}}{||\mathbf{w}_i||^2||\mathbf{w}_i||}$$

$$= \frac{\partial\mathcal{L}}{\partial\mathbf{v}_i}\frac{\beta_i}{||\mathbf{w}_i||} - \frac{\beta_i}{||\mathbf{w}_i||^2}\frac{\partial\mathcal{L}}{\partial\mathbf{v}_i}\frac{\mathbf{w}_i}{||\mathbf{w}_i||}\sum_j w_{ij}$$

$$= \frac{\beta_i}{||\mathbf{w}_i||}\partial_{v_{ij}}\mathcal{L} - \frac{\beta_i\mathbf{w}_i}{||\mathbf{w}_i||^2}\frac{\sum_j w_{ij}\partial_{v_{ij}}\mathcal{L}}{||\mathbf{w}_i||}$$

$$= \frac{\beta_i}{||\mathbf{w}_i||}\partial_{v_{ij}}\mathcal{L} - \frac{\beta_i}{||\mathbf{w}_i||^2}\mathbf{w}_i\partial_{\beta_i}\mathcal{L}. \tag{13}$$

Therefore, we have derived equation (7) (which is equation (13)).

## 4.5 Robustness to Weight Fluctuations

The decoupling of the weight matrix (*i.e.*, $\mathbf{v}_i = \beta_i\frac{\mathbf{w}_i}{||\mathbf{w}_i||}$) introduces a robustness to weights fluctuation. During learning, the distribution of the weights for each layer tends to remain more stable in NSM compared to StNN. See for instance Figure 4, where in the top row the evolution of weights distribution of the third layer ($\mathbf{W}_3$) is shown for the NSM and the StNN, respectively. It is apparent that the

distribution of NSM weights is more narrow and remains concentrated around its mean (low variance). On the other hand, the variance of the weight distribution in larger in the StNN. The same results are illustrated in the two bottom panels where the mean of the weights of the third layer over training is subtracted from the mean of the initial weights. We observe that the NSM is more robust and the mean remains almost steady (left panel) in comparison to StNN. The same phenomenon is observed also in the case of standard deviation (right panel), where the NSM's standard deviation increases slightly in comparison to StNNs.

Figure 4: Evolution of $W_3$ (*i.e.*, weights of the third layer) weight distributions during learning, normalized to initial values (top row). In the NSM, the scale of the weights is controlled by the factors $\beta_i$. This renders the weights during learning more stable (left panel, top row) compared to the sigmoid neural network (right panel, top row), which tends to grow at a faster rate. The mean of NSM remains close to zero (black line, bottom left panel) in comparison to the mean of the StNN (yellow line). Similar to the mean, the variance of NSM (black line, bottom right panel) grows slower and remains smaller than that of StNN (yellow line, bottom right panel).

## 4.6 Training NSMs with BinConcrete

This section details how the NSM can be trained using the BinConcrete distribution instead of propagating gradients through the activation probability function (see main text and SI 4.2). In the forward pass, the probability is computed using equation (3) and then passed to the BinConcrete [30] given by the following equation

$$X = \sigma\Big(\frac{L + \log(\alpha)}{\lambda}\Big), \tag{14}$$

where $\alpha$ is the probability we have already computed, $\sigma$ is the sigmoid function, $L$ is the Logistic distribution $(\log(U) - \log(1 - U)$, where $U$ is the uniform distribution in the $[0, 1]$) and $\lambda$ is the temperature term. In our experiments, we assume that $\lambda = 1$. In the backward pass, the gradients are computed through equation (14) instead of equation (3).

### 4.7  N-MNIST

The N-MNIST data set uses the same digits as contained in MNIST [39]. The digits were presented to an event-based camera that detects temporal contrast (ATIS), and their output was recorded. The data set consists of binary files, each containing the information of a single digit. Each file contains four arrays of equal length describing: the $x$ coordinate and $y$ coordinates of an event, the polarity (on or off) and the timestamp of the event. For this network, only the positive polarity events were extracted. Ten $34 \times 34$ frames of zeros were created for each digit and the maximum timestamp was divided by 10 to obtain the frame length. For each digit, an entry in the frame corresponding to the $x$ and $y$ coordinates of the events extracted inside the designated frame time was changed from 0 to 1. This was repeated for each of the ten frames. Test error results were obtained averaging test errors across the last 5 epochs and over $X$ separate runs with different seed values.

### 4.8  Simulation Details

The source code for this work is written in Python and Pytorch [41] and it is available online under the GPL license at `https://github.com/nmi-lab/neural_sampling_machines`. We ran all the experiments on two machines:

1. A Ryzen ThreadRipper with $64$GB physical memory running Arch Linux, Python 3.7.4, Pytorch 1.2.0 and GCC 9.1.0, equipped with three Nvidia GeForce GTX 1080 Ti GPUs.

2. A Intel i7 with $64$GB physical memory running Arch Linux, Python 3.7.3, Pytorch 1.0.1, and GCC 8.2.1, equipped with two Nvidia GeForce RTX 2080 Ti GPUs.

### 4.9  MNIST, EMNIST, NMNIST Neural Networks

Table 6: Convolutional neural network used for MNIST, EMNIST, NMNIST data sets.

| Layer Type | # Channels | $x, y$ dimension |
|---|---|---|
| Raw Input | 1 | 28 |
| $5 \times 5$ Conv | 32 | 24 |
| $2 \times 2$ Max Pooling (stride 2) | 32 | 12 |
| $5 \times 5$ Conv | 64 | 8 |
| $2 \times 2$ Max Pooling (stride 2) | 64 | 4 |
| $1024 \times 512$ FC | 1024 | 1 |
| Softmax output | 10 | 1 |

## 4.10 DVS Gestures Neural Network

Table 7: All convolutional neural network used for the DVS Gestures dataset.

| Layer Type | # Channels & Dimensions | |
|---|---|---|
| Input (ON events) | 6 | $64 \times 64$ |
| $3 \times 3$ Conv | 96 | $64 \times 64$ |
| $3 \times 3$ Conv | 96 | $64 \times 64$ |
| $3 \times 3$ Conv | 96 | $64 \times 64$ |
| $2 \times 2$ Max Pooling (stride 2) | 96 | $32 \times 32$ |
| $3 \times 3$ Conv | 192 | $32 \times 32$ |
| $3 \times 3$ Conv | 192 | $32 \times 32$ |
| $3 \times 3$ Conv | 192 | $32 \times 32$ |
| $2 \times 2$ Max Pooling (stride 2) | 192 | $16 \times 16$ |
| $3 \times 3$ Conv | 256 | $16 \times 16$ |
| $3 \times 3$ Conv | 256 | $16 \times 16$ |
| $3 \times 3$ Conv | 256 | $16 \times 16$ |
| $2 \times 2$ Max Pooling (stride 2) | 256 | $8 \times 8$ |
| $3 \times 3$ Conv | 256 | $8 \times 8$ |
| $1 \times 1$ Conv | 256 | $8 \times 8$ |
| $1 \times 1$ Conv | 256 | $8 \times 8$ |
| Global average pool | 256 | $1 \times 1$ |
| Softmax | 11 | $1 \times 1$ |

## 4.11 Weights Statistics for MNIST Classification

In this section we provide some statistics on the weights of the convolutional neural network used in the MNIST classification task. The network architecture is given in SI 4.9 and the results of the classification task are given in Table 2 in main text. First, we provide the histogram of the weights after training on the MNIST data set for three different types of networks. An NSM network, an NSM trained using the BinConcrete distribution and a deterministic network with sigmoid function as non-linearity. For more details about the networks see the main text, and SI 4.9 and 4.6. Histograms are illustrated in Figure 5. Then we measured the expected value of the weights for each layer and for each network as well as the mean gradients of the weights. Those results are shown in Figures 6 and 7, respectively. Finally, we show the angles between the gradients of NSM weights and NSM trained with BinConcrete (yellow) and NSM and Deterministic (orange) in Figure 8. These results indicate that the NSM and the NSM with BinConcrete express similar behavior during training. On the other hand, the deterministic network has larger weights and develops larger gradient steps (blue lines in Figures 6 and 7).

Figure 5: Histograms of Weights on MNIST Classification. The weights of four layers, convolutional layers 1 and 2 and fully connected layers 1 and 2 (see SI 4.9) for three different neural networks, NSM (yellow), NSM trained with Concrete Distribution (red, see SI 4.6), and Deterministic one with sigmoid as non-lineariry.

Figure 6: Mean of Weights on MNIST Classification. Mean weights of the four layers of the neural network used in main text for MNIST classification. Convolutional layers 1 and 2 and fully connected layers 1 and 2 (see SI 4.9). Weights of three different neural networks are presented here, NSM (yellow), NSM trained with BinConcrete Distribution (red, see SI 4.6), and a Deterministic one with sigmoid as non-linearity (blue).

Figure 7: Mean of Weights Gradients on MNIST Classification. Mean of weights gradients of the four layers of the neural network used in main text for MNIST classification. Convolutional layers 1 and 2 and fully connected layers 1 and 2 (see SI 4.9). Weights of three different neural networks are presented here, NSM (yellow), NSM trained with BinConcrete Distribution (red, see SI 4.6), and a Deterministic one with sigmoid as non-linearity (blue).

Figure 8: Angles (cosine similarity) of Gradients on MNIST Classification. The angles of gradients of weights of three networks are compared with each other. Four layers, convolutional layers 1 and 2 and fully connected layers 1 and 2 (see SI 4.9) are shown in this figure. Two cases are illustrated in this figure, (i) NSM against NSM trained with BinConcrete (yellow, see SI 4.6), (ii) NSM against a Deterministic network with sigmoid as non-linearity (red).