[Reviews · NeurIPS 2019]

Reviewer 1



**** I'm satisfied to accept this submission in light of the author's rebuttal, and have been inclined to increase my score to 7. **** Originality This work is an interesting work that unifies multiple perspectives and approaches, drawing from the state-of-the-art in binary neural networks, stochastic neural networks, neuromorphic approaches, deep neural networks, and weight normalization. Related work is well cited across these fields, and the approach is unique in this reviewer's knowledge. Quality The quality of this work is adequate, though there are a couple of simple errors in the text (misspelling in Figure 1, missing sections in the supplementary material, lack of explanation of some abbreviations such as W_3 and S2M). Overall, the text and derivation is done with high quality, and the tricks used in the derivation are called out to adequately describe the steps to the reader. The conclusions stand on their own, and quality of insight is needed to bridge stochastic neural networks, multiplicative weights, and weight normalization. The work could use more difficult datasets, though, to emphasize these results. The stochastic neural network literature uses more simple comparison datasets than are typically used in the deep neural network literature, as more basic questions of stability, trainability, and accuracy are still being addressed in a way that has been solved in other neural net subfields. Nonetheless, it can be difficult to know if results found on MNIST, CIFAR, and neuromorphic device datasets translate meaningfully to real-world domains. It can be helpful to try it on a truly challenging task to examine where the method breaks down. Additionally, there is some forcing in the derivation. The lack of an effective learning rule is not addressed in this work; smart initialization allows the experiments to run effectively even with a biased learning rule, while a truly well-matched learning rule would be ideal. Similarly, while one can train on the erf function as an activation function, and it even has a presence in particular neuron types [1], it is a fairly unusual function. On a positive note, this reviewer wishes to applaud the authors for the analysis of multiple noise types, which is useful as a hardware implementation of NSMs can have a variety of characteristics. Additionally, the diversity of experiments across datasets and neuromorphic sensor tests to validate the model are indeed substantial. Clarity This work contains acceptable clarity, going into detail about architectures, the derivation, the datasets, and the results. However, the derivation could be more clearly motivated by establishing earlier the aim and approach of the neural sampling machine formulation, as there are numerous branches of the derivation (multiplicative vs. additive noise, Gaussian vs. Poisson noise, the addition of an offset parameter to the noise). Establishing earlier the approach could set that structure up for the reader to more easily follow. Significance This result is significant to the neuromorphic engineering community, which can build devices that efficiently exploit many properties of neural sampling machines. Additionally, there are interesting implications for online learning or continual learning, especially in-device, and the work beckons future research to establish a more efficient learning algorithm for this formulation of neuron. This manuscript has the strong possibility to inspire other future work. [1] Jug, F., Lengler, J., Krautz, C., & Steger, A. (2012). Spiking networks and their rate-based equivalents: does it make sense to use Siegert neurons?. Swiss Society for Neuroscience.

Reviewer 2



Summary ------- This work combines stochastic neural networks with binarised neural networks. They introduce neural sampling machines (NSMs), which are neural networks with additive and/or multiplicative noise on the pre-activations. The main focus of the paper is on NSMs with multiplicative noise, however, which exhibit the weight normalising effects. Gaussian and Bernoulli noise sources are considered and training is done by backpropagating through the probability function of the binary activations, rather than using some variant of the REINFORCE algorithm. Comments -------- *originality*: The contributions of this paper are to the best of my knowledge original. The idea of self-normalisation, however, has been introduced with Self-Normalizing Neural Networks (SNNs) (Klambauer et al., 2017), which makes the title somewhat confusing, as well as the usage of SNN for stochastic neural network. *quality*: If I understood the purpose of binarised networks correctly, they are mainly interesting for porting networks to low-end hardware. Therefore, I would not expect binarised networks to outperform regular networks. This paper does claim to have binarised networks that are better than their non-binarised counterpart. From the current experiments, it is not clear to me where exactly the improvements comes from. Therefore, I think it would be interesting to additionally compare: - with standard binarised networks to assess improvements due to normalisation effect + stochasticity/learning rule - with normalised binarised networks to assess improvements due to normalisation effect - with normalised stochastic neural networks to assess improvements/deterioration due to binarisation. Am I correct to say that with the initialisation for CIFAR 10 (line 231), the NSM is practically a fine-tuned version of the deterministic model that you compare to? If yes, this would be a rather unfair comparison. Apart from the above and the typos listed below, this is a clean piece of work. - line 110: Shouldn't the probability of of a neuron firing be $P(z_i = 1 \mid z) = 1 - \Phi(u_i \mid z)$? - lines 136 and 142: the bias term should not appear in these equations for multiplicative noise, see line 118. - equation 7: The second term contains a derivative w.r.t. $\beta_i$, shouldn't this be w.r.t. $v_{ij}$? *clarity*: Overall, the paper is well written and easy to read. There are some points, however, where rewriting or adding more details might be useful to improve understanding and reproducability (might be partly resolved by code release): - line 122: "where $\beta$ is a parameter to be determined later" is rather uninformative. I would opt for something like "where $\beta$ models the effects due to the noise, $\xi$". - line 129: How important is this term $a_i$? Can't $\beta_i$ be controlled enough by tweaking variance (with Gaussian noise) or probability of success (with Bernoulli noise)? - line 170: Why are the gradients through the probability function a biased estimate of the gradient of the loss? - line 174: If you rely on automatic differentiation, does this imply that you have to compute both activations and probabilities in the forward pass? - line 187: What is this hardware implementation? Doesn't it suffice to test on software level? - line 191: What is "standard root-mean-square gradient back-propagation" in this context? From the text, it appears that the authors use a softmax layer with cross-entropy loss (quite confusing to also mention the negative log-likelihood loss, which is only an implementation detail in the end). So I assume it has nothing to do with the root mean squared error. It is also not clear how the data was split into training/validation and test data. The tables mention that test error was computed over 100 samples, but it is not clear how these samples were chosen. *significance*: Having powerful methods enabled in low-end hardware or even in hardware implementations will probably become increasingly important in a worl of "smart" tools. Stochasticity has already proven useful for uncertainty estimation, regularisation, etc. This paper effectively enables these tools in binarised networks, which are much easier to implement in hardware. The benefit of inherent normalisation with this method, makes this an especially interesting approach.

Reviewer 3



-------------------------------- Post-rebuttal comments -------------------------------- I want to thank the authors for their rebuttal. I am glad that we agree that implying that the presented architecture is intrinsically particularly well-suited to address the challenges presented by continual learning is a stretch, and I appreciate and agree with the clarification that the NSMs are on the other hand a good fit for online learning. I also want to thank them for the useful comparison using the STE on the NSM architecture, demonstrating that their proposed training procedure is indeed more effective. ------------------------------ Review before rebuttal ------------------------------ - Originality: The connection between binary stochastic activation and normalization seems novel. - Quality: The work seems of good quality - Clarity: The paper is generally well-written and clearly presented. One aspect though that is not motivated at all is the connection between continual learning, which they mention several times, including in the abstract. The authors do not explicitly address continual learning in the main body of the paper. They do not test their architecture on continual learning benchmarks, neither they seriously motivate what type of mechanism in their architecture would be useful for addressing the challenges of continual learning, such as catastrophic forgetting or transfer learning. - Significance: This is an interesting contribution proposing that binary stochastic activations confer an architecture self-normalization properties akin to weight and batch normalization. However, the statements implying that this mechanism might be inherently useful for continual learning is confusing and calls into question parts of this work and its presentation.

[Author Response · NeurIPS 2019]

**The camera-ready version of the manuscript will be modified with all the changes and new results we describe below. Thank you all reviewers for your in-depth comments.**

**Reviewer 1** We fixed the typos and abbreviations indicated by reviewer 1 in the text. S2M are Stochastic Sampling Machines, corrected. We added results on CIFAR100 using the same CIFAR10 network (test error rates for BNSM: $34.85\%$, dCNN: $34.37\%$). Regarding the comments about "smart" initialization on CIFAR10 experiment, we ran experiments without that and we achieved similar results (CIFAR10 with initialization: Test error rate$10.2\%$, without: $9.94\%$). We restructured the text such that the NSM is better and earlier introduced in the main text, add derivation details and provide the experimental results. Previously unexplained terms (*e.g.*, the offset parameter to the noise) are now elaborated. We have not attempted NSMs in a (deep) RL framework as we are not aware of convincing theoretical/practical work on successful deep RL with binary neural networks. On the other hand, using the RL framework to de-bias learning rules in StNN is an active field of study and explored in MuProp [18] and closely related REBAR. With MuProp, the number of passes in the network for each gradient step scales with the number of layers, which does not scale to the larger networks used here. Backpropagating through the probability function yields a good result while being straightforward to implement in software and hardware. See response to reviewer 3 concerning the learning rule bias. We improved the SI by elaborating on Figure 3, fixing missing sections (4.3) and detailing the DVS gestures dataset and preprocessing.

**Reviewer 2** Abbreviation SNN (Stochastic Neural Network) was changed to StNN. We suggest "Inherent Weight Normalization in Stochastic Neural Networks" as a less confusing title. Klambauer et al. in 2017 indeed introduced self-normalization. Our work is different in terms of objective and results. Klambauer et al. construct *mathematically* an activation function with which outputs are normalized in non-binary, deterministic networks. *Weight* Normalization emerges in the NSM unit from the multiplicative noise. That the central limit theorem in such units leads to weight normalization in the Salimans & Kingma sense is original, and establishes a one-of-a-kind connection between exploiting the physics of hardware systems and recent deep learning techniques, while achieving reasonable accuracy on classification tasks. This is highly significant for the devices community, as it implies a simple circuit (threshold operations and crossbars) that can exploit (rather than mitigate) device non-idealities such as stochasticity. Many challenges remain to achieve hardware NSMs, but we believe this is a seminal step. Reviewer suggested three more experiments. So, we tested a binary (sign non-linearity) deterministic network (BD), the same with weight normalization (wBD), a stochastic network (noisy rectifier) (SN) on MNIST (all these networks were trained using a Straight-Through Estimator (STE)) and a deterministic Binary Network (BN) trained using erf in the backward pass giving these results: BD: $3.11\%$, wBD: $2.72\%$, SN: $2.05\%$, BN:$1.10\%$, NSM: $0.70\%$. Regarding NSM's initialization process, we ran a more experiments using the same allconvnet implementation without pre-trained weight initialization, which achieved similar results (CIFAR10 test error rate $9.95\%$). We corrected the probability of a neuron firing (line 110, line 108 was correct). We removed the bias term from equations in lines 136 and 142 since it was a mistake (pointed out by the reviewer 2). Regarding Eq. (7), our calculations confirm the derivative w.r.t. $\beta_i$. See also Eq. (3) in [47]. The details of the derivation will be added to the SI. We agree with the expression "where $\beta$ models the effects due to the noise, $\xi$". $a_i$ (line 129) is critical for controlling $\beta_i$. We did initially investigate whether $\beta_i$ can be controlled by tweaking $\sigma$ (Gaussian noise) or $p$ (Bernouilli noise) instead. The former case would work theoretically, but in the latter case, the free parameter is $0 < p < 1$. The bounds translate to bounds on $a_i$ that are too restrictive for training the networks. Note that $\beta_i$ is a trainable parameter and it (or $a_i$) must be stored anyway, and using $a_i$ is process-independent and so preferable. The gradients (line 170) are biased because they are estimated using switching probability rather than the realization Eq (3). The NSM is compatible with the binary case of Concrete relaxations [31] with fixed temperature, which are unbiased w.r.t the continuous network. To assess the bias, we trained the NSM using BinConcrete units [31] on MNIST (test error rate: $0.78\%$). We observed that the angles between two gradients (current and BinConcrete) are close. The training indeed relies on automatic differentiation (line 174) but probabilities are only computed for the gradients. NSMs propagate forward binary states only. A computational graph will detail this in the SI. We removed the confusing line 187 regarding the hardware implementation, as the NSM non-volatile memory (NVM) crossbar implementation is not documented here (see subsection 1.3 in the main text). The expression in line 191 was changed to "NSM was trained using back-propagation and a softmax layer with cross-entropy loss and mini-batches of size 100". The dataset splitting in training/testing is standard (50K/10K for MNIST, and 50K/10K for CIFAR). By "error was computed over 100 samples" we meant that for testing, each mini-batch is run 100 times (MC samples) with different seeds. Classification is made on the average over the MC samples. We'll change the wording.

**Reviewer 3:** Our original statement about continual learning was wrong. We replaced it by "online learning". NSM theoretically supports online learning since it implements weight normalization, which is not based on batches [47] (online learning capability is relevant to physical neural networks since batching is not possible). NSM training on hardware is out of scope here, but it is in principle compatible with analog NVMs: The gradient of the erf is a Gaussian, and so gradients w.r.t. $v$ (see main text) can be sampled with noise in the device. The gradient w.r.t. $\beta$ are more challenging, but only one per neuron needs to be computed, which can be done in peripheral CMOS. We agree with the reviewer that the present paper would benefit from a quantitative comparison with STE. To this end we, trained an NSM ($\sim 1\%$) and an NSM ($\sim 3\%$) using a STE on MNIST. The proposed NSM was more accurate.

[Meta-Review · NeurIPS 2019]

The author rebuttal went a long way towards satisfying the reviewers and all of them have recommended acceptance after discussions. The authors should go through the suggestions given by the reviewers (esp. about harder dataset experiments, better derivations, better ablations) and incorporate as much as possible in the camera-ready version of this paper.